# Microencapsulation as a Tool for the Formulation of Functional Foods: The Phytosterols’ Case Study

**DOI:** 10.3390/foods9040470

**Published:** 2020-04-09

**Authors:** Roberta Tolve, Nazarena Cela, Nicola Condelli, Maria Di Cairano, Marisa C. Caruso, Fernanda Galgano

**Affiliations:** School of Agricultural, Forestry, Food and Environmental Science, University of Basilicata, 85100 Potenza, Italy; roberta.tolve@unibas.it (R.T.); nazarena.cela@unibas.it (N.C.); nicola.condelli@unibas.it (N.C.); maria.dicairano@unibas.it (M.D.C.); marisa.caruso@unibas.it (M.C.C.)

**Keywords:** phytosterols, functional foods, cholesterol reduction, microencapsulation, legislation

## Abstract

Hypercholesterolemia, which is an increase in total and low-density lipoprotein (LDL) serum cholesterol, is an important risk factor for the development of cardiovascular diseases. Lifestyle modifications underpin any action plan for reducing serum cholesterol. Phytosterols are natural compounds belonging to the triterpenes family. Thanks to their structural analogy with cholesterol, phytosterols have the ability to reduce serum LDL-cholesterol levels. Phytosterols are used to enrich or fortify a broad spectrum of food products. Like unsaturated fatty acids and cholesterol, phytosterols are easily oxidized. Microencapsulation could be a useful tool to overcome this and other drawbacks linked to the use of phytosterols in food fortification. In this review, in addition to explaining the phytosterols’ mechanisms of action, a focus on the use of free and encapsulated phytosterols for the formulation of functional foods, taking also into account both technological and legislative issues, is given.

## 1. Introduction

Phytosterols, also called plant sterols, are natural compounds belonging to the triterpene family. They are structurally similar to cholesterol, except for the presence of a methyl or ethyl group at the C-24 carbon atom of their side chain. On the other hand, phytostanols are the saturated form of phytosterols, differing from them for the absence of the double bond. Both phytosterols and stanols can be esterified with fatty acids of vegetable oil origin. The resulting esters are liquid or semi-liquid materials, having comparable chemical and physical properties to edible fats and oils, enabling supplementation of various processed foods. While cholesterol is a fundamental component of animal cell membranes, phytosterols play important metabolic functions in plant cells. These compounds perform a key role in the regulation of membrane permeability and fluidity. Moreover, phytosterols are involved in embryogenesis and are precursors of brassinosteroids [1]. However, they cannot be synthesized endogenously in humans and are derived solely from diet. The biosynthesis of these compounds occurs through the succession of 30 enzyme-catalyzed reactions. The whole synthesis process can be separated into two major stages: The mevalonate pathway and the enzymatic steps particular to sterols. The main precursor is squalene, synthesized from acetyl-CoA by a series of reactions. This compound is transformed into 2,3-oxidosqualene and this into cycloartenol. Then, a series of oxidative and demethylation reactions begin, for example, methylation at C24, dehydrogenation, and isomerization of the double bond. The main phytosterols, whose structure is reported in Figure 1 are: β-sitosterol, campesterol, stigmasterol, brassicasterol, campestanol, and β-sitostanol [2]. The importance of phytosterols is due to their action of reducing low-density lipoprotein (LDL) cholesterol levels, and a daily consumption of 2–3 g of phytosterols could decrease the LDL-cholesterol by 10–15% [3]. In addition, phytosterols have antiinflammatory, antidiabetic and anticarcinogenic actions and are involved in the prevention of colon, breast, and prostate cancer [4]. Phytosterols are considered GRAS (generally recognized as safe) and, due to their beneficial effects on human health, are widely used as functional food ingredients [5]. “Functional foods” is a relatively new term that dates back to the last century: It was introduced for the first time in Japan in the mid-1980s and it is used to describe food products enriched or fortified with natural components with a specific physiological preventive or health-promoting effect [6].

Considering that the content of phytosterols naturally present in foods does not reach the dose that should be taken to observe a reduction of the LDL serum cholesterol, the production of foods fortified with phytosterols is becoming increasingly important. However, the phytosterols’ incorporation into food products is challenging because of their susceptibility to oxidation, water insolubility, and chalky taste. These obstacles could be removed through the development and optimization of microencaspsulated phytosterols, obtained via the bioactive packaging in protective matrices [4,7,8]. Although in literature other reviews concerning the microencapsulation of bioactive compounds have been published [9,10,11], to our knowledge none about the phytosterols’ microencapsulation is available. Against this background, the purpose of this review was to provide a summary about the use of free and encapsulated phytosterols for the formulation of functional foods, taking into account both technological and legislative issues.

## 2. Effects and Action Mechanisms of Phytosterols

A high level of LDL cholesterol in the plasma, due to an incorrect diet, is connected to the onset of coronary diseases, which are one of the main causes of death and significantly also affects the health care cost. The reduction of cholesterol levels in plasma prevents many cardiovascular diseases. A way to achieve this objective, and avoid drugs’ use, could be the consumption of phytosterol-enriched/fortified foods. Although cholesterol and phytosterols have a similar structure, they are absorbed and metabolized by the human body with different efficiency. As a matter of fact, while the absorption of cholesterol is between 20% and 80%, the absorption of phytosterols is approximately 2–5%. As a consequence of this, the plasma levels of phytosterols are very low (0.10–0.14% of the cholesterol levels). Phytostanols are absorbed even less efficiently than the cholesterol and their blood levels are equal to one-tenth of those of phytosterols [12]. The phytosterols and cholesterol, which come through the diet or with the enterohepatic circulation in the small intestine, are incorporated into micelles, which, interacting with the apical membrane of the enterocytes, are incorporated into the intestinal cells [5]. The intestinal absorption of cholesterol and phytosterols/phytostanols occurs through a complex multistep process. Within the enterocyte, cholesterol is esterified by the enzyme AcilCoA:cholesterol acyl transferase-2 (ACAT2) and becomes part of the chylomicrons, lipoproteins rich in triglycerides that carry the dietary fats in the plasma. The low plasma concentrations of phytosterols could depend on their poor affinity for the ACAT2 enzyme, which would prevent their incorporation into the chylomicrons. The nonesterified cholesterol and phytosterols are transported back to the intestinal lumen and from there to the liver via the enterohepatic circulation. This step is extremely important since it counteracts the sterols’ accumulation into the enterocyte, preventing the formation of cytotoxic products of the lipid peroxidation. The leakage of cholesterol and phytosterols from the enterocyte is mediated by carriers of the “ATP-binding cassette transporters” or ABC transporters named ABCG5 and ABCG8 [13]. There are several mechanisms of action through which the phytosterols reduce the blood cholesterol level. The most important mechanism is the competition of phytosterols with cholesterol for incorporation into the mixed micelles in the intestinal chyme, because of the structural similarity of the two molecules. In this way, cholesterol remains not solubilized, its absorption is reduced, and it will be expelled through the feces [14]. Secondly, phytosterols interfere with the normal mechanisms responsible for maintaining intracellular cholesterol homeostasis. This effect depends on many factors, including the activity of the ABC transporters. In fact, phytosterols compete with cholesterol during its transport into enterocytes by means of the NPC1L1 (Niemann-Pick C1 Like 1) transport proteins. Once in the enterocytes, phytosterols are preferentially transported back out into the lumen by ABCG5 and G8 heterodimer complex. Phytosterols, which are incorporated into chylomicrons and enter into the circulatory system, are excreted from the liver into the bile by this ABCG5/G8 system (Figure 2) [14].

The reduced intestinal absorption of cholesterol (consequent to the modification of mixed micelles induced by phytosterols) can lead to an increase in both the synthesis of endogenous cholesterol and the uptake of lipoproteins in the liver. In fact, when the cholesterol concentration inside the cell goes down, the sterol regulatory element-binding proteins (SREBPs) activate the transcription of both the enzyme 3-Hydroxy-3-methylglutaryl-CoA reductase (HMGCR) and the receptor for LDL, causing an increase of the endogenous biosynthesis of cholesterol and of the LDL cells’ uptake, respectively. Both effects lead to an overall reduction in circulating cholesterol [15]. In contrast to the favorable effects of plant sterols on cholesterol serum levels, in some cases they are considered atherogenic, but only where there is a rare genetic disease called sitosterolemia (or phytosterolemia). Sitosterolemia is a rare autosomal recessive inherited disorder, resulting from a mutation in either the ABCG5 or the ABCG8 gene, in which phytosterols are over-absorbed and accumulate in the tissues, causing xanthomatosis and premature atherosclerotic diseases [16,17]. Before the first description of sitosterolemia, it was believed that plant sterols were not absorbed by the human body. In 1974, for the first time, it was diagnosed a case of sitosterolemia in the United States and, with subsequent studies, the mechanisms by which this disease is generated were also explained. Afterwards, it was estimated that the mean consumption of plant sterols is about 400 mg/day; healthy subjects absorb less than 5% (while plant stanols are absorbed only for 0.5%). In sitosterolemic patients it was found a high concentration of phytosterols in the plasma, ranging from 10 to 65 mg/dL, and also an increase in the levels of cholestanol, 5α-campestanol, and 5α-sitostanol, starting from their unsaturated precursors. The symptoms of this disease can be confused with cases of hypercholesterolemia or xanthomatosis, but it can be discriminated from these disorders because the only clinical sign of sitosterolemia is an abnormal hematological parameter. In fact, a sitosterolemic patient exhibits normal or slightly increased cholesterol levels. Furthermore, it is believed that extremely high levels of plant sterols cause stiffness of the membrane, rupture of red blood cells, and changes in platelet size, number, and function, which occasionally causes bleeding episodes. The known therapy for sitosterolemia consists of a diet low in plant sterols. Fortunately, this is a rare disease [18,19]. Another negative effect linked to the consumption of phytosterols is the reduction of levels of fat-soluble vitamins. In fact, as phytosterols reduce the absorption of cholesterol by a mechanism of competition with it for the incorporation into mixed micelles, there is also a lower absorption of other lipophilic compounds, such as β-carotene, α-carotene, and vitamin E (tocopherols). The effect of phytosterols and phytostanols on the absorption of fat-soluble vitamins has been known since the year 2000 when the first phytosterols-enriched foods present in the market were deeply studied. The commercialization approval for these foods was issued with the condition that the labeling state that “the product may not be nutritionally appropriate for certain sections of the population (pregnant and breastfeeding women and children under the age of five years)” and “the product should be used as part of a healthy diet, including regular consumption of fruit and vegetables (to help to maintain carotenoid levels)”.

The different studies led to the conclusion that there is a reduction of the level of about 24% of β-carotene and a reduction of about 9% for α-carotene, while there are no significant changes for retinol (vitamin A) and vitamin D. However, this negative effect can easily be countered by eating more colored fruits and vegetables [20].

## 3. Phytosterols as Natural Source or Added in Foods

The main sources of phytosterols are vegetable oils, nuts, legumes, fruits, and vegetables (Table 1). This is the reason why the Mediterranean diet, which is rich in plant foods, gives a big contribution to the intake of the right amount of phytosterols. It should be noted that particular oils, such as rice bran oil or wheat germ oil, can contain phytosterols in amounts up to 3200 mg/100 g. Nuts, instead, contain phytosterols in the range of 30–220 mg/100 g of product [21,22]. Phytosterols are one of the most important bioactive compounds for their positive effect on lowering blood cholesterol levels. Thanks to their capacity, they are added in different matrices. The first phytosterols-enriched food was marketed in Finland in 1995 where Benecol margarine was fortified with plant stanol fatty acid esters. Since then, many other functional foods enriched in phytosterols have been developed and are now available globally [23]. Matrices in which these bioactive compounds are added are different, generally lipid matrix but the most widely used are, for example, dairy products, spread, bakery products, and meat products, but also in low-lipid matrices, like orange juice [24]. There are several factors that must be taken into account when approaching the development of a new product.

As an initial step, it is necessary to verify, in the country where production and marketing must take place, which legislation regulates the new products and if the addition of certain bioactive compounds in a particular food matrix is allowed. From a technological point of view, it is necessary to: Evaluate the interaction between the functional compound and food matrix; determine the quantity of compound to be added; evaluate the possible effect on the product or on the compound of the different operations to which the product will be subjected; and validate the final product from a chemical, physical, and sensorial point of view, but also make in vivo tests to show that currently the product has positive effects on human health. In fact, sometimes it is not possible to add any bioactive compound in any food matrix, because these compounds could be incompatible, could develop anomalous odors or flavors, and/or could excessively change the organoleptic characteristics of the product. Moreover, the compounds could be damaged during the production process or they could be sensitive to the action of gastric juices in the gastrointestinal tract. One of the factors that must always be considered is the bioavailability of these compounds. That is, how much of this molecule is actually assimilated by the human body? Recently, Vaghini et al. [26] showed that the bioavailability of phytosterols is a function of the type of matrix in which they are inserted. The digestion process was simulated to see how much of this compound was transferred from the food matrix to the mixed micelles in the intestinal chyme. A greater bioavailability percentage means greater incorporation into the mixed micelles and, therefore, greater cholesterol displacement from the latter. As reported by Alvarez-Sala et al. [27], the presence of other substances used as ingredients in food formulation might have an effect on the solubility and bioavailability of phytosterols and it was also noted that the effectiveness of phytosterols on the reduction of LDL cholesterol levels was influenced by the type of food matrix (e.g., its nutritional composition, presence of additives or other bioactive compounds, etc.), by the number of portions per day, time of intake, if they are consumed as a snack or as a meal, their origin, etc. It emerged that, for example, milk and yogurt allow a greater reduction of LDL cholesterol and that their appropriate intake should be during meals and not at breakfast on an empty stomach and that the presence of other bioactive compounds could improve the action of phytosterols. In another study, the rheological and physical properties of a yogurt enriched with phytosterols during storage were evaluated [28]. The phytosterols’ dispersion was prepared using an oil/water emulsion (O/W) and was added to the milk. Subsequently, important parameters linked to the quality and the microstructure of yogurt were evaluated, such as pH, acidity, syneresis, firmness, and the viscosity of the enriched yogurt. As a result, viscosity and syneresis were lower, while firmness was greater than the control. Furthermore, the addition of phytosterols had a significant influence on acidity, while, as far as the sensory characteristics were concerned, no significant differences emerged between the phytosterols-enriched sample and the control, considering the consistency, appearance, flavor, and product’s overall acceptability. Moreover, it was noticed that the distribution of phytosterols in the sample was not homogeneous, tending to settle in a small layer of the sample.

In order to support the food industry and assure the correct use of claims on the nutritional labelling for food fortified with phytosterols, robust analytical techniques for the extraction and the quantification of these compounds are required [29]. The most widely used analytical methods to measure phytosterols in food products commonly consist of their alkaline saponification using potassium hydroxide or sodium hydroxide at concentrations ranging from 1 to 6 mol/L [29,30]. This step is followed by an organic solvent extraction using hexane, heptane, toluene, or petroleum ether [23,29,30,31,32]. The selection of the organic solvent is a crucial step influenced by several factors, such as the affinity with the target compounds, the availability, the safety, and its hydrophilicity [29]. Organic solvent extraction is then followed by derivatization into trimethylsilyl ether derivatives followed by separation and quantification by gas chromatography (GC) [30]. Both GC with flame ionization detection (GC-FID) and GC-mass spectrometry (GC-MS) have been widely used [31].

## 4. Stability of Phytosterols in Foods

As well as the issue mentioned so far, it is necessary to report that phytosterols are unstable at high temperatures and, being unsaturated lipid compounds, are very susceptible to oxidation. In fact, following the various production processes to which the various products are subjected, the phytosterols undergo transformation. Therefore, it is necessary to maintain the quality and safety of phytosterols-enriched foods during processing and storage. However, it is notable that phytosterols are more stable than mono-unsaturated fatty acids (e.g., oleic acid), because of steric hindrance in the ring structure. In any case, the factors that lead to the transformation of the phytosterols, causing their oxidation, are not only the high temperatures and the heating time, but also the composition of the matrix in which they are present. A suggestion is to add phytosterols in products that will not suffer, then perform heat treatments at temperatures above 100 °C [16]. Regarding the stability during processing and storage, the phytosterols and phytostanols are microbiologically inert. In fact, they resist in fermented products, for example, in yogurt and fermented milks, so the oxidative stability is one of the most important problems [16]. Following oxidation, the phytosterols are transformed into POPs (phytosterols oxidation products), which could exert toxic effects qualitatively similar to those of cholesterol oxidation products (COPs), for which it is clarified their link with the most common chronic diseases [33]. POPs’ production takes place following a sequence of successive reactions, from which are obtained primary compounds or hydroperoxides; secondary compounds, such as ketones, alcohols, and epoxides; and tertiary compounds like dimers, oligomers, and polymers. The most common POPs are keto-, epoxy-, and hydro-compounds, in particular 7-ketocampesterol, 7-keto-β-sitosterol, 7-β-OH-campesterol, and 7-β-OH -β-sitosterol. No evidence of genotoxic effects in vivo was observed. Numerous in vitro experiments showed that POPs exert cytotoxic effects but certainly five times less than the cholesterol oxidation products. The POPs have a potential pro-atherogenic effect and increase production of reactive oxygen species [34]. High blood levels of POPs increase the risk of developing cardiovascular disease. They exert cytotoxic effects comparable to those associated with COPs. POPs, in fact, lead to a reduction of the membrane integrity and reduction of mitochondrial dehydrogenase activity and lead to reduced cellular viability. While there are several studies on the COPs’ biological functions, the knowledge concerning POPs is still limited, which would require further investigation [35]. The various factors influencing the production of these compounds have also been evaluated. First of all, analyses were carried out on different groups of foods enriched with different concentrations of free phytosterols or phytosterols esters, for example, in butter, margarine, vegetable oils, milk, chocolate, fruit juices, and milk-based beverages [36]. It was seen that the factors involved in POPs’ production were: The amount of phytosterols added, the temperature and the heating time, the chemical form of phytosterols, and the food matrix in which they were added. High content of phytosterols leads to a higher POPs’ production, with the phytosterols as their precursors. The storage temperature has little influence on the formation of POPs in food, while the temperature and heating time is much more important. The influence of the matrix has been demonstrated by comparing the amount of POPs produced in rapeseed oil, margarine, and butter. The highest POPs’ content was found in butter followed by margarine and finally in rapeseed oil. According to Lin et al. [36], the daily dose of POPs should not exceed 47.7 mg, so it is extremely important to consider the content in phytosterols and their stability in functional foods in order to evaluate their safety [37]. In fact, it was carried out a study using dark chocolate enriched with phytosterol esters and was assessed its oxidative stability during 5 months of storage. The samples considered were unfortified chocolates used as a control, chocolates enriched with phytosterols, and chocolates containing phytosterols and ascorbic acid and α-tocopherol as antioxidants. Samples were stored at 20 and 30 °C for 5 months. Chocolates stored at 20 °C had the maximum production of hydroperoxides, the first oxidation products, after 60 days of storage. After this period, the hydroperoxides’ content decreased because of their transformation into the secondary oxidation products. At 30 °C, the maximum level of hydroperoxides occurred after 30 days. Chocolates containing phytosterols were more oxidized than the control. Another note is that the inclusion of antioxidants, in addition to phytosterols, was not effective in reducing oxidation because there were no differences in the content of hydroperoxides between the chocolates that contained antioxidants and those formulated only with phytosterols. Probably, the ascorbic acid and α-tocopherol concentration used (0.90 mg/100 g) was not effective to reduce the chocolate oxidation. The production of hydroperoxides accelerated the oxidation of phytosterols, reducing their beneficial activity in chocolates, as the hydroperoxides and free radicals generated catalyzed the phytosterols oxidation [38]. Rudzińska et al. [39] studied the phytosterols’ susceptibility in margarine. In detail, they stored margarine for 18 weeks at two different temperatures, 4 °C and 20 °C. The addition of phytosterols in free form led to their decrease at the end of storage, and also there was an increase of the respective oxidation products. Moreover, the storage temperature significantly affected POPs’ production: At 20 °C, phytosterols were oxidized about 1.5 times faster than at refrigeration temperature (4 °C). Therefore, storage conditions are important to avoid the formation of POPs with negative health effects. The stability of the phytosterols has been studied in different food matrices, for example, on fruit juices, as in the study carried out by González-Larena et al. [37]. It is to be pointed out that the type of packaging used for storage may also affect the stability of a food product enriched with phytosterols. As shown by Semeniuc et al. [40], in order to better preserve the characteristics of a yogurt beverage enriched with phytosterols, it is possible to operate on the packaging in order to reduce the contact with light. In the specific case, the suitable packaging material was a mini plastic bottle with triple black layer and the completely opaque intermediate layer. In this way, it was possible to prevent lipid oxidation because contact with light was avoided and photodegradation processes were prevented. Moreover, to date, to ensure that the phytosterols do not undergo transformations during the production processes or during their conservation, a technique that allows to protect the bioactive compounds added to the food matrices could be used. This technique is microencapsulation.

## 5. Micro/Nanoencapsulation of Phytosterols

Microencapsulation could be defined as a process by which microparticles or droplets are surrounded by a coating, or embedded in a homogeneous or heterogeneous matrix, to give small capsules with many useful properties [41]. This process, suitable to entrap active agents within a carrier material, could be a beneficial tool to enhance the delivery of bioactive compounds and living cells into foods. The carrier material, called shell or wall, used in food products has to be food grade and able to form a barrier between the active agent and its surrounding environment. One of the most important reasons for the encapsulation of bioactive compounds is to provide greater stability during processing as well as in final products. Another benefit of the encapsulation process is the reduction of the evaporation and the degradation of active volatile compounds, including aroma components, and the prevention of reaction between the bioactive compound and other components in food products including oxygen and water [42]. As well as to reduce the oxidation of the phytosterols, with the POPs’ production, the microencapsulation of these compounds could be suitable to overcame other drawbacks, like their chalky taste and water insolubility [4]. There are several techniques that have been developed for microencapsulation of phytosterols, such as spray-drying, freeze-drying, complex coacervation, and liposome [43,44,45,46]. All these techniques, in combination with the different shell materials, strongly impacted the functionality of phytosterols [47]. When the microencapsulation of bioactive compounds is carried out with the purpose to enrich a food product, it is important to evaluate the cost and the solvent that may be used. As an example, Leong et al. [48] evaluated the possibility to produce phytosterols nanodispersions, with particle sizes of 50 nm, using the emulsification-evaporation technique with different types of organic phases (hexane, isopropyl alcohol, ethanol, and acetone). Hexane allowed obtaining the smallest particle size. Moreover, it emerged from this study that higher homogenization pressure and a higher organic:aqueous phase ratio resulted in significantly larger phytosterol nanoparticles. Also, Auweter et al. [49] have drawn up a protocol for phytosterols’ microencapsulation that is involved in the use of organic solvent, for the dissolution of the active compound, followed by a dispersion of the obtained mixture into an aqueous medium containing the shell polymers. In this latter research, before dehydrating the solution via spray-dryer technology, the removal of the solvent was necessary. In fact, the major disadvantage of this protocol is the use of solvents, which could negatively affect both the product healthiness and production costs. For this reason, this review will be focused only on the studies that do not consider the use of organic solvent (Table 2). Chen et al. [43] co-encapsulated fish oil, phytosterol esters and limonene in whey protein isolate, and sodium caseinate shell materials using spray-drying technique. These authors obtained good quality microcapsules, with higher retention of eicosapentaenoic and docosahexaenoic acids, known as EPA and DHA, and better flavor/odor profile. The accelerated storage for 7 days at 45 °C and 30% relative humidity under saturated oxygen showed that the encapsulated actives were characterized by a higher protection from oxidation than the non-encapsulated ones. The same authors had also shown that the dehydration made using the freeze-drying method was not advantageous compared with spray-drying to produce the fish oil emulsion co-encapsulated with phytosterol and limonene microcapsules, as also reported by other researchers [50]. This could be due to the fact that the freeze-dried samples in their original form have limited usage for food application, and a grinding process of samples into powder format is required. This step produces a more “unsealed” surface on the particles [43]. About that, Alvim et al. [51] remarked how the use of low temperatures for the dehydration of a liquid system containing microcapsules may represent an advantageous alternative compared to other processes, because of the mild processing conditions, the good levels of retention of volatile components, and possible large-scale production. In detail, these authors investigated the production of lipid microparticles containing phytosterols by spray-chilling technology. The microencapsulation via spray-chilling differs from the freeze-dryer technology, because it is based on the spraying, in a chilled chamber, of a lipid mixture containing the active compound, in this case, phytosterols. The protection obtained through the microencapsulation may also involve the increase in phytosterols’ bioavailability. This has been shown by Ling and Lin [52], who evaluated the in vitro release of oven-dried microencapsulated phytosterols, contained in kenaf seed oil, by using alginate with high methoxy pectin and chitosan as shell materials. Similar results were also reported by Lim and Nyam [53], who produced spray-dried microcapsules of kenaf seed oil using carboxymethyl-cellulose, maltodextrin, and soy lecithin as shell materials. Beyond the bioavailability of the microencapsulated bioactive compound, it is very important to evaluate both its shelf life and retention. With this purpose, Khalid et al. [54] studied the microencapsulation of β-sitosterol in an O/W emulsion obtained by microchannels’ emulsification. These authors obtained a phytosterols’ retention of 80% when they stored the samples for 30 days at 4 and 25 °C. Slightly different results were reported by Comunian et al. [55], who evaluated the effect of different combinations of shell materials and crosslinkers’ agent on the co-encapsulation of echium oil and β-sitosterol by complex coacervation. The phytosterols’ retention was 73.78 and 70.74% when stored for 30 days at 37 °C depending upon the presence or the absence of sinapic acid as crosslinking, respectively. Other researchers demonstrated that the concentration of phytosterols contained in kenaf seed oil is stable when stored at 65 °C for 24 days. In detail, in this study the impact of the accelerated storage conditions on microencapsulated kenaf seed oil, obtained by spray-dryer using sodium caseinate and maltodextrin as shell polymers, was evaluated [56]. According to Di Battista et al. [57], in order to produce a stable emulsion with lipophilic bioactive compounds it may be necessary to achieve high temperatures (above the melting point of phytosterols, or about 136 °C). However, this step could negatively affect the oxidative stability of phytosterols. That was reported by Tolve et al. [4], who solubilized phytosterols in soybean oil prior, to produce an O/W emulsion with selected shell materials: Whey protein isolate, inulin, and chitosan. The emulsion dehydration was obtained with spray-dryer but, unexpectedly, the peroxide values of the obtained microcapsules were relatively high even just after the production. These results have been attributed to the combination of high temperatures and high shear mixing used in the production of the emulsions and to the phytosterols which, at low concentration, might have acted as a pro-oxidant, as also suggested by Winkler and Warner [58]. In order to overcome this drawback, the same research group carried out the phytosterols’ microencapsulation using inulin, chitosan, and whey protein isolate as shell material, by formulating aqueous suspensions with the addition of surfactant [59].

## 6. Use of Microencapsulated Phytosterols for Functional Foods’ Production

The studies about the microencapsulation of phytosterols brought back here do not include their incorporation into food matrices. However, this step is fundamental for the production of new fortified food, because it allows both the evaluation of the stability and the acceptability of the final product. In this regard, Bagherpour et al. [60] added encapsulated β-sitosterol in butter, since saturated fat consumption increases cardiovascular disease. For the encapsulation of β-sitosterol, it was used nanostructured lipid carriers, prepared by hot melt homogenization method. Beta-sitosterol-loaded lipid nanocarriers, characterized by a particle size of about 165 nm, showed good stability during three months’ storage period. Moreover, it was also shown by this study that the addition of β-sitosterol-loaded lipid nanocarriers in the concentration of 1 mg/g does not alter the texture and the organoleptic characteristics of the product. It is interesting to note that, for the formulation of food fortified with microcapsules, it is important to not exceed the particle size of 20–30 μm in order to not affect the texture and the mouth feel of the final product. In particular, in respect of phytosterols, it was demonstrated that an average particle size of about 25 μm could enhance their incorporation into the intestinal micellar phase, improving, in this way, phytosterols’ bioavailability [4]. Also, Comunian et al. [61] developed a functional yogurt containing encapsulated phytosterols. Phytosterols were co-encapsulated with echium oil using sinapic acid as a crosslinking agent and a mixture of arabic gum and gelatin as shell material. Both yogurt fortified with microencapsulated phytosterols and without active ingredients were produced and the yogurt containing microcapsules in the concentration of 0.2 mg/g did not show a significant difference in physicochemical, rheological, and sensorial properties with the control. Recently, Tolve et al. [62] developed functional dark chocolates enriched with three different levels of microencapsulated phytosterols: 0.5, 1, and 1.5 mg/g. The obtained chocolates were characterized by a particle size distribution lower than 30 μm and were stable from a chemical point of view. Moreover, results showed that, although an increase in microencapsulated phytosterols’ concentration affected the chocolate sensory attributes, all the samples thoroughly reached the threshold of acceptability. Although the latter study represents an applicative example of the phytosterols’ microencapsulation for the production of fortified food products, it should be noted that, to date, it is not allowed the inclusion of these compounds in chocolate products. For this reason, in-depth knowledge of the phytosterols’ legislation is recommended before proceeding with the formulation of a food product fortified with free or encapsulated phytosterols.

## 7. Legislation

Manufacturers who decide to produce a functional food enriched with phytosterols must respect the specific regulations in accordance with member state law where the product will be marketed. In fact, there are several legislations that regulate the marketing of phytosterols-fortified/enriched products. Regulations rule the choice of matrix, the serving sizes, the claims to be reported on the label, and allowable maximum daily doses. These regulations vary between countries (Table 3). In Europe, following the diffusion of the first phytosterols-enriched food in Finland, the European Parliament and Council enacted the Regulation (EC) 258/97 on Novel Food Regulations [63]. These regulations shall apply to novel foods and ingredients that have not been significantly consumed by humans before 15 May 1997. Sterol esters were approved as novel food ingredients in 2000 and were authorized to be used in margarine to reduce plasma cholesterol levels. In 2004 the Regulation was extended to other matrices, for example milk-based beverages, yogurt, and other fermented milk-type drinks, soy drinks, fat spreads, salad dressings, spicy sauces, and cheese-type products with less than 12% of fat. The producers of these functional foods must comply with the regulations regarding the claims to be reported on the label. In fact, as regards in the European Union, there exists the Regulation (EC) 608/2004 [64] concerning the labeling of the product or food ingredients added with phytosterols, phytosterols esters, phytostanols, and/or phytostanol esters, modified then from Regulation (UE) 718/2013 [65]. Food products which provide a daily intake of at least 2 g sterols/stanols in the label must be indicated by the following indications: The words “addition of plant sterols/vegetable stanols”, the content of phytosterols/phytostanols and their esters in percentage or in grams per 100 g of foodstuff, and it must be specified that the product is not intended for people who do not need to check the level of cholesterol. Moreover, it should be noted that patients who follow a cholesterol-lowering treatment should consume the product only under medical supervision and it must be stated clearly and legibly that the product is inadequate for pregnant or nursing women or for children under 5 years of age. In addition, it must indicate that the product fortified with phytosterols should be taken as part of a balanced diet that involves regular consumption of fruit and vegetables to help maintain carotenoid levels and the portion of the food product or the food ingredient concerned must be specified in order to avoid consumption of more than 3 g/day. In Europe, according to the Commission decision of 31 March 2004 [66], the addition of phytosterols is authorized only in matrices, such as yellow fat spreads, salad dressings, milk-type products, fermented milk-type products, soy-based drinks, and cheese-like products. Instead, in other countries like the USA, Canada, Australia, New Zealand, Japan, and other Asian countries, these functional ingredients can also be added in other food matrices. As for the USA, the Food and Drug Administration (FDA) has approved phytosterols as GRAS compounds. Plant sterols were also approved to be added in other matrices, such as dairy products, bakery products, cereals, oils and salad dressings, and juices. The claims that must be reported on the label, if the product contains at least 0.5 g per serving of phytosterols eaten with meals or snacks for a daily total intake of 2 g, are “helps maintain normal cholesterol level” or “may reduce the risk of heart disease”. In Australia, it is necessary to refer to the Food Standards Australia New Zealand [67] which, in 2001, approved the use of plant sterols in matrices, such as breakfast cereals, oils, milk, yogurt, and even cheese. The claim to be clearly marked on the label of the food product containing at least 0.8 g of phytosterols for a daily intake of 2 g is “reduces blood cholesterol”. In Canada, the addition of phytosterols was approved only in 2010, following the numerous scientific studies that demonstrated and validated the beneficial effects on health. Besides the food products in which it is possible to add plant sterols, like the usual margarine, yogurt, and condiments, it is possible to fortify also vegetables and fruit juices. Particular is the claim that is used: “Plant sterols help reduce/lower cholesterol. High cholesterol is a risk factor for heart disease.” Foods are eligible for health claims if they: Contain a minimum level equivalent to 0.65 g of free plant sterols or stanols per reference amount and per serving of stated size, contain 100 mg or less of cholesterol per 100 g of food, and other characteristics established by the specific Canadian Regulation [68]. The term “functional food” was coined in Japan. In detail, in 1991, the Ministry of Health, Labor, and Welfare enacted FOSHU (Food for Specified Health Uses) [65] as a regulatory system for approval of foods that have beneficial effects on human health. For the first time, functional foods were considered as a category distinct from conventional and medical foods. The first phytosterols-enriched food was oil. The claim added to the label of these products is: “Good for those concerned about serum cholesterol”. In Japan, the legislation that provides for the addition of these claims is much more severe. In fact, the approval process is very rigorous [69]. In other Asian countries, such as China, functional foods enter in the category of healthy foods that comprise other types of products. These, in order to be defined as such, must be fully characterized, manufactured under the current good manufacturing practice (GMP), and supported by safety data, quality assessment, and efficacy study. It must also be specified that the functional product is not a substitute for medicines. In 2007, a Regulation was issued similar to that on European Novel Food, in which plant sterols were the first real ingredients to be added to the foods regulated by this law. In China, to date, only a milk drink enriched with plant sterols has been approved and launched on the Chinese market. Even in Taiwan, as in China, functional foods are part of a category of healthy foods that help reduce the risk of disease. A particular feature of this country is that health claims are called maintenance claims. Seven maintenance claims have been approved including “regulating blood lipids” [70,71]. In South Korea, functional foods are those products that have undergone transformations to improve and maintain human health through the consumption of functional ingredients. There are 37 categories of bioactive foods and ingredients and among these there is the category “products containing phytosterols”. In addition to red yeast rice and soy protein, plant sterols are considered ingredients that lead to the reduction of cardiovascular diseases and, therefore, can be added both in food and in dietary supplements [68,69,70,71]. Phytosterols-enriched/fortified foods have also been approved in other Asian countries, such as Malaysia, Indonesia, the Philippines, and Singapore, and in general the matrices that are used to add phytosterols are mostly milk-based beverages. As far as South America, Brazil is one of the few countries with specific legislation that regulates functional foods. There are various matrices where it is possible to add phytosterols, such as vegetable margarine, yogurt, and, recently, milk. The claim used is “helps maintain healthy level of cholesterol when associated with a healthy diet and lifestyle”. It is possible to find some products containing phytosterols in Argentina, Chile, Colombia, Ecuador, and also in Mexico [72].

## 8. Conclusions

Since many scientific evidences support the beneficial effects of phytosterols on human health, their use for the formulation of functional food is increasing. On this basis, it is important to protect phytosterols against less suitable conditions (moisture, oxidants, light, and temperature, among others) in order to improve their stability and the shelf life of the final product. It should be further considered that phytosterols, like unsaturated fatty acids and cholesterol, are easily oxidized. So, in order to enhance phytosterols’ applications in food systems, these compounds may be encapsulated by developing different techniques. Indeed, there is not a unique technique which allows us to produce encapsulated phytosterols. The inherent characteristics of shell materials and their interaction with the phytosterols, the microcapsules’ stability during processing and storage, the release rate, and the interaction with the carrier food must be assessed during the selection of the technique. There is a growing number of studies about phytosterols’ microencapsulation, but very few works on the interaction between the microencapsulated phytosterols and the food. Thus, more efforts are required to allow the possibility to use microencapsulated phytosterols for the formulation of functional foods, considering that several studies have already suggested and proposed their application.

## Figures and Tables

**Figure 1 foods-09-00470-f001:**
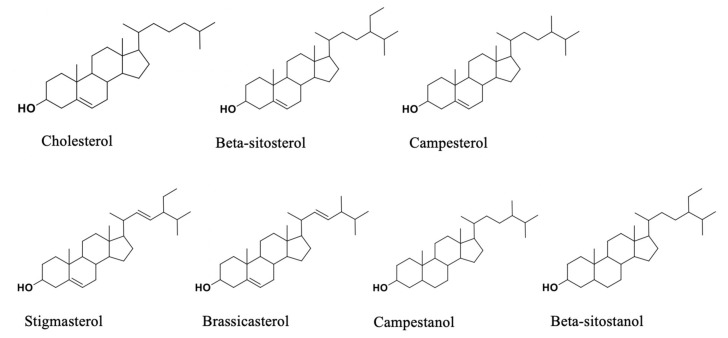
Cholesterol and some phytosterols’ structures.

**Figure 2 foods-09-00470-f002:**
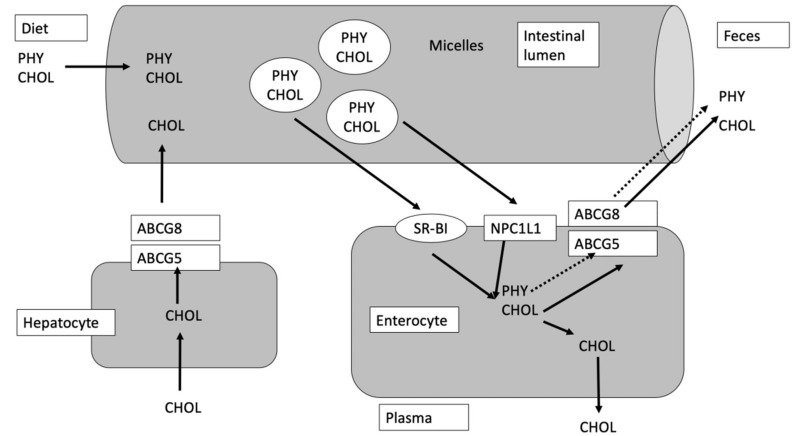
Transport mechanism of CHOL (cholesterol) and PHY (phytosterols), modified from Jesch and Carr [5]. SR-BI is the scavenger receptor class B type I, NPC1L1 is the Neiman-Pick C1 Like 1 and ABCG5 and ABCG8 are the transporters ATP binding cassette (ABC) G5 and G8, respectively.

**Table 1 foods-09-00470-t001:** Phytosterols’ food sources and their content.

Phytosterol Food Sources	Phytosterols Content (mg/kg of Fresh Weight)	Reference
Oils		
Corn	7150–9520	[23]
Olive	1140–1150	[25]
Palm	490–610	[25]
Peanut	1670–2290	[25]
Rice bran	10,550	[23]
Soybean	2210–3280	[23]
Sunflower	2030–3280	[25]
Vegetables		
Broccoli	367–390	[25]
Carrot	153–160	[25]
Cauliflower	310–400	[25]
Onion	84–93	[25]
Potato	38–73	[25]
Tomato	47–148	[25]
Fruits		
Apple	130–183	[25]
Banana	116–161	[25]
Grapes	40–200	[25]
Orange	228–240	[25]
Nuts		
Almond	1380–1430	[25]
Peanuts	600–1608	[25]
Cereals		
Barley	720–801	[25]
Buckwheat	963–1980	[25]
Corn	662–1205	[25]
Oats	350–491	[25]
Rye	707–1134	[25]
Wheat	447–830	[25]

**Table 2 foods-09-00470-t002:** Overview of microencapsulated phytosterols and their application.

Core Material	Shell Material	Encapsulation Technique	Food Inclusion	Principal Outcomes	Reference
Fish oil, phytosterol esters and limonene	Whey protein isolate and sodium caseinate	Spray drying	N.R.	Higher protection from oxidation than non-encapsulated phytosterols	[43]
Phytosterols mixture	Lipid mixture of low trans hydrogenated vegetable fats and stearic acid	Spray chilling	N.R.	Good quality microcapsules with a mean diameter varied between 13.8 and 32.2 μm	[51]
Kenaf seed oil containing phytosterols	Alginate with high methoxy pectin and chitosan	Oven-dried	N.R.	Increase in phytosterols bioavailability evaluated through in vitro release	[52]
Kenaf seed oil containing phytosterols	Carboxymethyl-cellulose, maltodextrin and soy lecithin	Spray drying	N.R.	Increase in phytosterols bioavailability evaluated through in vitro release	[53]
Beta-sitosterol and γ-oryzanol	Medium chain triglycerice oil	O/W microchannel emulsification	N.R.	Phytosterols retention ranged from 50 to 80%, according to the use of tween 20 or decaglycerol monolaurate as surfactant agent, when stored for 30 days at 4 and 25 °C	[54]
Beta-sitosterol and echium oil	Arabic gum, cashew gum	Complex coacervation	N.R.	Phytosterols retention ranged from 70.74 to 73.78% depending upon the absence or the use of sinapic acid as crosslinking when stored for 30 days at 37 °C	[55]
Kenaf seed oil containing phytosterols	Sodium caseinate and of maltodextrin	Spray drying	N.R.	Phytosterols concentration was stable when microcapsules were stored at 65 °C for 24 days	[56]
Phytosterol mixture	Arabic gum, maltodextrin	Spray drying	N.R.	The microcapsules particle size was lower than 25 μm, which is required to ensure the phytosterols inclusion in the intestinal micellar phase	[57]
Phytosterol mixture	Whey protein isolate, inulin and chitosan	O/W emulsion + spray drying	N.R.	Unexpected, the peroxide values of the obtained microcapsules were relatively high even just after the production	[4]
Phytosterol mixture	Whey protein isolate, inulin and chitosan	Spray drying	N.R.	Possibility to scale up the production of microcapsules without affect their features using a laboratory dryer or a spray dryer for semi- technical production	[59]
Beta-sitosterol	Lipid mixture of Precirol and Miglyol	Hot melt homogenization method	Butter	Beta-sitosterol loaded lipid nanocarriers, showed good stability during three months’ storage period Moreover, the use of this technique does not alter the texture and the organoleptic characteristics of the product	[60]
Echium oil and beta-sitosterol	Arabic gum and gelatin	Complex coacervation	Yogurt	Yogurt containing microcapsules did not show a significant difference in terms of physicochemical, rheological and sensorial properties with respect to control	[61]
Phytosterols mixture	Whey protein isolate	Spray drying	Dark chocolates	No matter the microencapsulated phytosterols concentration, fortified dark chocolate was widely accepted by consumers	[62]

N.R., not reported.

**Table 3 foods-09-00470-t003:** Legislation on phytosterols-enriched foods in different countries.

Country	Current Legislation	Health Claim
European Union (EU)	Novel Food Regulations (EC 258/97)	“Plant sterols (stanols) have been shown to lower/reduce blood cholesterols. High cholesterol is a risk factor in the development of coronary disease” “Plant sterols/stanols contribute to the maintenance of normal blood cholesterol levels”
United States of America	GRAS notification and self-GRAS regulation; Dietary Supplement Health and Education ACT (DSHEA)	“Helps maintain normal cholesterol levels”; “May reduce the risk of heart disease”
Australia and New Zealand	Novel Food Standard; Food Standards Australia New Zealand (FSANZ)	“Reduces blood cholesterol”
Canada	Part B, Division 28 (Novel Foods) of the Food and Drug Regulations	“Plant sterols help reduce/lower cholesterol. High cholesterol is a risk factor for heart disease”
Japan	Food for Specified Health Uses (FOSHU)	“Good for those concerned about serum cholesterol” “Good for those having relatively high serum cholesterol and triglycerides with mild obesity”
China	State Food and Drug Administration (SFDA)	“This product is not a substitute for medicine”
Taiwan	Health Food Control Act	“Regulating blood lipids”; “An animal study shows that consumption of this product may help lower blood total cholesterol”
South Korea	Korea Health Functional Food Act (HFFA) by Korean Food and Drug Agency (KFDA)	“Phytosterols may reduce the risk of coronary heart disease”
Malaysia	Food Safety and Quality Division under Malaysian Regulations of the Food Act	“Helps lower or reduce cholesterol”
Indonesia	Indonesian National Agency for Drug and Food Control (NADFC)	“May reduce the risk of coronary heart disease”
Thailand	Thai Food and Drug Administration	“May help lower cholesterol”
Philippines	Philippine Food Fortification Act	“This product contains plant sterols that help lower cholesterol”
Singapore	Implemented by Agri-Food and Veterinary Authority with the Health Promotion Board	“Plant sterols/stanols have been shown to lower/reduce blood cholesterol. High blood cholesterol is a risk factor in the development of coronary heart disease”; “Intended exclusively for people who want to lower their blood cholesterol level”
Brazil	National Health Surveillance Agency	“Helps to maintain healthy level of cholesterol when associated with a healthy diet and life style”
Mexico	Mexican General Health Law	“Proven to reduce cholesterol”

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
