# Peer review of "Microencapsulation as a Tool for the Formulation of Functional Foods: The Phytosterols’ Case Study"

_foods, 2020, doi:10.3390/foods9040470_

Round 1

Reviewer 1 Report

The subject and overall content of the paper is very interesting with rich information about the mechanism of action of phystosterols, challenges associated with the fortification of foods with phytosterols and how encapsulation is used to tackle those challenges. The review also highlights gap to be filled by further research and important considerations for developing shelf stable foods fortified with phytosterols. Wording should be reconsidered and some sentences rephrased. I have made a number of suggestions in the detailed comments (see below).

Page 1 Line 10: Hypercholesterolemia, which the increase.... :   add “is” between which and  an increase or reword the sentence  
Page 1 Line 14:   I would delete  “Due to this attitude”, and  consider  for this reason,  phytosterols …

Page 1 Line 16-20:  Consider rewording reword the sentence.

Page 1 Line 31-32 …. Fluidity.  Moreover, phytosterols are involved in embryogenesis and are precursors of brassinosteroids [1]. However, they cannot be synthesized endogenously in humans and are derived solely from diet.

Page 1 Line 40:   … reducing LDL cholesterol levels and a daily consumption of 2–3 g of phytosterols could decrease the LDL by….
Page 1 Line 41: …. have also anti-inflammatory… delete “also”
Page 2 Line 50: …. the content ….foods, do not… replace “do” by “does”
Page 2 Line 55-56:  start with Against this background, the purpose of this review is to provide a summary…. foods, taking into account…… The mechanisms of action of  phytosterols is also discussed.  

Page 3 Line 83: .. intestinal chime because of the structural similarity of the two moelcules.

Page 4 Line 139 -140: … Finland in 1995 where  Benecol margarine was fortified  with plant stanol…

Page 4 Lines 144: … that must be taken into account when approaching the development of a new product.

Page 5 Lines 160-161:. The digestion process was simulated…

Page 5 Lines 174-176:  consider rewording and be concise

Page 8 Line 270: …particle size. Moreover, it emerged from this study that higher …

Page 14 to the end…Line ?

Page 18Line 88-89: 89 ..there are growing  number of studies about  phytosterols microencapsulation, but very few works  focused on…

Author Response

Foods. Ref. No.:  Foods-767231

Authors: Roberta Tolve, Nazarena Cela, Nicola Condelli, Maria Di Cairano, Marisa C. Caruso and Fernanda Galgano

Title: Microencapsulation as a tool for the formulation of functional foods: the phytosterols case study.

Dear Reviewer,

The authors are grateful to the reviewers for their careful read and constructive comments. As suggested, we used the version you provided in the e-mail.  Moreover, in order to facilitate the further editor and reviewers’ activity, we used the "Track Changes" function in 
Microsoft Word. All the proposed comments have been taken into account in the revised version. Additionally, other minor modifications, easily identifiable using the “Track Changes”, were carried out.

Authors’ answers to Reviewer 1 Comments  

Line

Reviewer’s Comment

Answer to Reviewer’s comments

Page 1 Line 10

Hypercholesterolemia, which the increase.... :   add “is” between which and  an increase or reword the sentence  

As suggested by the Referee, this correction has been made.

Page 1 Line 14

I would delete “Due to this attitude”, and  consider  for this reason,  phytosterols …

As suggested by the Referee, this correction has been made.

Page 1 Line 16-20

Consider rewording reword the sentence.

As suggested by the Referee, this correction has been made.

Page 1 Line 31-32

Fluidity.  Moreover, phytosterols are involved in embryogenesis and are precursors of brassinosteroids [1]. However, they cannot be synthesized endogenously in humans and are derived solely from diet.

As suggested by the Referee, this correction has been made.

Page 1 Line 40

reducing LDL cholesterol levels and a daily consumption of 2–3 g of phytosterols could decrease the LDL by

As suggested by the Referee, this correction has been made.

Page 1 Line 41

have also anti-inflammatory… delete “also”

As suggested by the Referee, this correction has been made.

Page 2 Line 50:

the content ….foods, do not… replace “do” by “does”

As suggested by the Referee, this correction has been made.

Page 2 Line 55-56: 

start with Against this background, the purpose of this review is to provide a summary…. foods, taking into account…… The mechanisms of action of  phytosterols is also discussed.  

As suggested by the Referee, this correction has been made.

Page 3 Line 83:

intestinal chime because of the structural similarity of the two moelcules.

As suggested by the Referee, this correction has been made.

Page 4 Line 139 -140:

Finland in 1995 where  Benecol margarine was fortified  with plant stanol…

As suggested by the Referee, this correction has been made.

Page 4 Lines 144:

that must be taken into account when approaching the development of a new product.

As suggested by the Referee, this correction has been made.

Page 5 Lines 160-161:.

The digestion process was simulated…

As suggested by the Referee, this correction has been made.

Page 5 Lines 174-176: 

consider rewording and be concise

As suggested by the Referee, this correction has been made.

Page 8 Line 270:

particle size. Moreover, it emerged from this study that higher …

As suggested by the Referee, this correction has been made.

Page 14 to the end

Line ?

Maybe a problem during the formatting occurred. Anyway, in this revision line numbering is present throughout the text.

Page 18Line 88-89: 89

there are growing  number of studies about  phytosterols microencapsulation, but very few works  focused on…

As suggested by the Referee, this correction has been made.

Reviewer 2 Report

The manuscript is a bibliographic review on the microencapsulation of bioactive compounds, particularly the phytosterols as a means to be preserved them against oxidation. It is well written and understandable. Nevertheless, and over all, I miss data on amounts of phytosterols. Mainly, the amounts added to the foodstuffs and amounts that are used to microencapsulate.

Also, I miss more references about other similar papers; for example:

Microencapsulation of bioactives for food applications

MI Dias, ICFR Ferreira, MF Barreiro –

Food & function, 2015

Natural phytochemicals and probiotics as bioactive ingredients for functional foods: Extraction, biochemistry and protected-delivery technologies

Beatriz Vieira da Silva , Joao C.M. Barreira, M. Beatriz P.P. Oliveira

Trends in Food Science & Technology 50 (2016) 144e158

Other comments to consider:

Figure 1. Revise the name:   Stigmasterol in place of Sigmasterol

Table 1. It would be better to show data from different authors showing a range of the data found, and not one data, as the values for each oil fluctuate depending on many factors.

Line 189. the next sentence sounds contradictory: “However, even under severe conditions, such as prolonged frying or too high temperatures, sterols could oxidize”

Line 218…."The highest POPs content was found in butter"

It has not sense, since in butter the major sterol is cholesterol, there are no phytosterols therefore.

Line 230-231: ….“it would have been useful to add the antioxidants in much higher quantities”

More quantities do not imply more effectiveness, even less quantity can be more effective. The most important would be to study the exactly quantity at which the antioxidant is effective.

It could be of great interest if you indicate the quantities of phytosterols used in the different published papers:

For example:

Line 325 …. “Bagherpour et al. [50] have encapsulated b-sitosterol in butter, since saturated fat consumption increases cardiovascular disease”….

Line 337….yogurt fortified with microencapsulated phytosterols

Line 339… Tolve et al. [52] developed different functional dark chocolates enriched with microencapsulated phytosterols.

Quantities?

In relation to legislation, Lines 20-28 is an unique and very long sentence, please divide.

Table 3. Title?. Why is not include the quantities to be allowed in each case?

It is not mentioned methods for phytosterol determination

Author Response

Foods. Ref. No.:  Foods-767231

Authors: Roberta Tolve, Nazarena Cela, Nicola Condelli, Maria Di Cairano, Marisa C. Caruso and Fernanda Galgano

Title: Microencapsulation as a tool for the formulation of functional foods: the phytosterols case study.

Dear Reviewer,

The authors are grateful to the reviewers for their careful read and constructive comments. As suggested, we used the version you provided in the e-mail.  Moreover, in order to facilitate the further editor and reviewers’ activity, we used the "Track Changes" function in 
Microsoft Word. All the proposed comments have been taken into account in the revised version. Additionally, other minor modifications, easily identifiable using the “Track Changes”, were carried out.

Authors’ answers to Reviewer 2 Comments  

Line

Reviewer’s Comment

Answer to Reviewer’s comments

Also, I miss more references about other similar papers; for example:

Microencapsulation of bioactives for food applications

MI Dias, ICFR Ferreira, MF Barreiro –

Food & function, 2015

Natural phytochemicals and probiotics as bioactive ingredients for functional foods: Extraction, biochemistry and protected-delivery technologies

Beatriz Vieira da Silva , Joao C.M. Barreira, M. Beatriz P.P. Oliveira Trends in Food Science & Technology 50 (2016) 144e158

As suggested by the Referee, other similar papers have been added. Please, see the reference from 7 to 11:

-Dias et al., (2015);

-da Silva et al. (2016);

-Abbas et al., (2012);

-Gonçalves et al., (2016); 

-Timilsena et al. (2017).

Figure 1.

Revise the name:   Stigmasterol in place of Sigmasterol

As suggested by the Referee, this correction has been made.

Table 1.

It would be better to show data from different authors showing a range of the data found, and not one data, as the values for each oil fluctuate depending on many factors.

As suggested by the Referee, the range has been added.

Line 189.

the next sentence sounds contradictory: “However, even under severe conditions, such as prolonged frying or too high temperatures, sterols could oxidize”

The reviewer is right. Although the meaning of the sentence is that severe temperature conditions can bring to the phytosterols oxidization, the meaning could be misleading. For this reason, we decide to delete the sentence.

Line 218

"The highest POPs content was found in butter" It has not sense, since in butter the major sterol is cholesterol, there are no phytosterols therefore.

This sentence referred to a paper in which phytosterols have been added to different food product, including butter. For this reason, the POP’s amount has been evaluated.

Line 230-231:

“it would have been useful to add the antioxidants in much higher quantities” More quantities do not imply more effectiveness, even less quantity can be more effective. The most important would be to study the exactly quantity at which the antioxidant is effective.

As suggested by the reviewer the amount of antioxidant has been added.

Line 325 ….

It could be of great interest if you indicate the quantities of phytosterols used in the different published papers: Bagherpour et al. [50] have encapsulated b-sitosterol in butter, since saturated fat consumption increases cardiovascular disease”…. Quantities?

As suggested by the Referee, the range has been added.

Line 337

yogurt fortified with microencapsulated phytosterols. Quantities?

As suggested by the Referee, the range has been added.

Line 339

Tolve et al. [52] developed different functional dark chocolates enriched with microencapsulated phytosterols. Quantities?

As suggested by the Referee, the range has been added.

Legislation Lines 20-28

is an unique and very long sentence, please divide.

As suggested by the Referee, the range has been added.

Table 3

Title?.

Title: Maybe a problem during the formatting occurred. Anyway, in this revision line numbering is present throughout the text.

Table 3

Why is not include the quantities to be allowed in each case?

Unfortunately, in this moment, we didn't get a chance to found the quantities of phytosterols allowed in each country and, for this reason, we decide to add the founded information in the main text instead in the table 3.

It is not mentioned methods for phytosterol determination

The reviewer is right. We added the method commonly used for phytosterols extraction and determination (pag. 6, line 253-263)

Round 2

Reviewer 2 Report

I agree with all the corrections made by the authors and consider that the manuscript is much better. Therefore, in my opinion it is ready to be published.